# Histopathology of Skeletal Muscle in a Distal Motor Neuropathy Associated with a Mutant CCT5 Subunit: Clues for Future Developments to Improve Differential Diagnosis and Personalized Therapy

**DOI:** 10.3390/biology12050641

**Published:** 2023-04-23

**Authors:** Federica Scalia, Everly Conway de Macario, Giuseppe Bonaventura, Francesco Cappello, Alberto J. L. Macario

**Affiliations:** 1Department of Biomedicine, Neurosciences and Advanced Diagnostics (BiND), University of Palermo (UNIPA), 90127 Palermo, Italy; giuseppe.bonaventura@unipa.it (G.B.); francesco.cappello@unipa.it (F.C.); 2Euro-Mediterranean Institute of Science and Technology (IEMEST), 90139 Palermo, Italy; econwaydemacario@som.umaryland.edu (E.C.d.M.); ajlmacario@som.umaryland.edu (A.J.L.M.); 3Department of Microbiology and Immunology, School of Medicine, University of Maryland at Baltimore-Institute of Marine and Environmental Technology (IMET), Baltimore, MD 21202, USA

**Keywords:** chaperone system, molecular chaperone, chaperonopathies, distal neuropathies, human CCT, CCT5 mutation, apical domain, skeletal muscle, muscle pathology, protein aggregates, desmin, immunohistochemistry, immunofluorescence, molecular dynamics simulations

## Abstract

**Simple Summary:**

The morphological alterations of the tissues caused by genetic chaperonopathies are scarcely known. Therefore, the histopathological identification of these diseases is a frustrating task, likely to end in misdiagnosis. Consequently, genetic chaperonopathies appear to be rare but, with more histopathological information and better diagnostic accuracy, assessment of their incidence will certainly reveal that they are not that rare and will most likely unveil many cases with mild clinical manifestations. However, progress in this area is impeded by the lack of illustrative publications describing the histopathology of genetic chaperonopathies. As a contribution to remedy this situation, in this brief review, we summarize data from the few previous publications on this topic that are available. The bulk of the data discussed comes from our earlier report on a chaperonopathy associated with a missense mutation in the gene encoding the CCT5 subunit of the chaperonin CCT. The case was diagnosed as a distal motor neuropathy of early onset, and we carried out a detailed histopathological analysis of the abnormalities present in the patient’s striated muscle. Here, we discuss a few of those results that we think offer the best outlook for further exploration to elucidate pathogenic mechanisms and provide clues for diagnosis.

**Abstract:**

Genetic chaperonopathies are rare but, because of misdiagnosis, there are probably more cases than those that are recorded in the literature and databases. This occurs because practitioners are generally unaware of the existence and/or the symptoms and signs of chaperonopathies. It is necessary to educate the medical community about these diseases and, with research, to unveil their mechanisms. The structure and functions of various chaperones in vitro have been studied, but information on the impact of mutant chaperones in humans, in vivo, is scarce. Here, we present a succinct review of the most salient abnormalities of skeletal muscle, based on our earlier report of a patient who carried a mutation in the chaperonin CCT5 subunit and suffered from a distal motor neuropathy of early onset. We discuss our results in relation to the very few other published pertinent reports we were able to find. A complex picture of multiple muscle-tissue abnormalities was evident, with signs of atrophy, apoptosis, and abnormally low levels and atypical distribution patterns of some components of muscle and the chaperone system. In-silico analysis predicts that the mutation affects CCT5 in a way that could interfere with the recognition and handling of substrate. Thus, it is possible that some of the abnormalities are the direct consequence of defective chaperoning, but others may be indirectly related to defective chaperoning or caused by other different pathogenic pathways. Biochemical, and molecular biologic and genetic analyses should now help in understanding the mechanisms underpinning the histologic abnormalities and, thus, provide clues to facilitate diagnosis and guide the development of therapeutic tools.

## 1. Introduction

The chief components of the chaperone system of an organism are the molecular chaperones, which are typically cytoprotective but which, if abnormal, can become etiopathogenic factors and cause diseases known as chaperonopathies [1,2]. These diseases can be genetic or acquired, with the former caused by mutations in chaperone genes that code for defective chaperone proteins with pathogenic potential. Examples are mutations of the chaperone CCT5 subunit that are associated with distal neuropathies [3,4,5,6]. For instance, the His147Arg CCT5 mutation is associated with a mutilating sensory distal neuropathy, whereas the Leu224Val mutation is associated with an early onset distal motor neuropathy. The discrepancy in patients’ phenotypes is a challenge for researchers who wish to understand the molecular mechanisms that cause the phenotypic differences, including both those that cause abnormalities of nervous tissue and those that are involved in the pathogenic pathways affecting other tissues, such as muscle. Likewise, the differences between the two mutations in clinical and pathological manifestations is a conundrum for clinicians who must make a correct differential diagnosis. This problem may be serious, because these varieties of genetic chaperonopathies may be more frequent than currently believed, and the tissue abnormalities present in them have not been fully characterized. Histological results have only been provided for the Leu224Val CCT5 mutation, as reported in our previous work [6].

CCT5 is the fifth of the eight subunits that form the two rings of the chaperonin-containing TCP-1 (CCT) hetero-oligomeric complex, which is also called the TCP-1 ring complex (TRiC) [7]. The CCT complex is a constitutively expressed chaperonin of Group II that assists in the folding of a least 15% of the cytosolic proteins, including cytoskeletal proteins, e.g., actin and tubulin, and proteins of the cell cycle, using energy from ATP hydrolysis [7]. CCT5, like the other seven subunits of the CCT complex, consists of three domains: (a) the apical domain involved in the binding of the client proteins; (b) the equatorial domain bearing the ATPase necessary for generating energy from ATP hydrolysis and interacting with the neighboring subunits; and (c) the intermediate domain connecting the apical domain to the equatorial domain and transferring, along the entire molecule, the conformational changes necessary for function [8,9].

Here, we review recent findings in muscle tissue of a CCT5 chaperonopathy characterized by the Leu224Val mutation accompanied by distal motor neuropathy of early onset [4,6].

The aim of this article is to highlight various morphological abnormalities that we have found by applying histological techniques in the skeletal muscle of a patient, findings we think deserve further investigation using biochemical and molecular methods. We combine our data with the scarce data reported in the literature to propose a working hypothesis for investigating molecular mechanisms and to stimulate research to elucidate these molecular mechanisms, particularly the role of defective chaperoning by the CCT complex that comprises the mutant CCT5 subunit. Clarification of these mechanisms will provide clues that will help pathologists decide which molecules and structures should be targeted for identification, quantification, and mapping, with the purpose of characterizing the histopathological abnormalities that are distinctive for each chaperonopathy. This information will be key in achieving accurate differential diagnosis and, thereby, improving patient care.

## 2. Histomorphological Abnormalities Associated with CCT Chaperonopathy

The main morphological abnormalities of the muscle tissue detected with standard histological procedures are reported in Table 1 (illustrative images may be found in reference [6]).

*Hematoxylin—eosin and Alcian–PAS staining.* Histological staining of skeletal muscle tissue from the patient and healthy control tissue demonstrated atrophy of the diseased tissue, as shown by the hematoxylin–eosin staining [6].

An abundant extracellular matrix separates the fibers that are smaller than those in normal muscle and have irregular morphology (e.g., they are sigmoidal in the longitudinal section and rounded in cross-section) and swollen nuclei. Most nuclei are subsarcolemmal, but a few are paracentric. The relationship between cell morphology and CCT5 alterations have been discussed in the literature. For example, it was demonstrated that the in vivo silencing of CCT subunits led to changes in the morphology of a human colon carcinoma cell line, and that those cells treated with siRNA for CCT5 were narrow and elongated [10]. Studies on skeletal muscle in a zebrafish model harboring a mutation in the CCT5 subunit showed enlarged sarcoplasm and an overall reduction of myofibrils [11]. Accordingly, it is possible that some of the abnormalities of myofibers morphology we observed in our patient were the consequence of the CCT5-Leu224Val mutation.

*Apoptosis.* Information on the involvement of the CCT complex or its subunits in skeletal muscle apoptosis is scarce. The programmed cell death protein 5 (PDCD5), a pro-apoptotic factor that regulates the function of the CCT complex [12], and the gelsolin protein, a calcium-regulated protein cleaved by caspase-3 when it is not bound to the CCT complex [13], have roles in the induction of caspase-dependent cell death. The CCT interactome is variable, according to the tissue type and the status of the cell (e.g., stressed or not). The interactions between CCT subunits, CCT subunits and ATP or ADP molecules, the CCT complex and client proteins, and the CCT complex and chaperones and/or co-chaperons are regulated by finely tuned allosteric movements of subunit domains and by chemical and physical conditions (e.g., pH and electrostatic potential) outside and inside the chaperonin complex [14,15]. Alterations of one of these parameters—for instance, due to amino acid substitutions—may lead to malfunction, e.g., some proteins may not be properly folded, and others may be too strongly bonded or sequestered by the chaperonin. In addition, other proteins that are found to be altered in the affected skeletal muscle tissue, such as desmin and αB-crystallin, as discussed later, may contribute to apoptosis [16,17].

## 3. Molecular Abnormalities Associated with CCT Chaperonopathy

Molecular abnormalities detected with immunohistochemical methods and immunofluorescence are listed in Table 2 (illustrative images may be found in reference [6]).

*CCT subunits*. Studies on the in vivo expression and localization of CCT subunits in healthy human skeletal muscle tissue have not been reported. Our group recently found, through ex vivo analysis using immunofluorescence (IF), the CCT5 subunit in the cytoplasm, nuclei, and sarcolemma of healthy human skeletal muscle tissue [6]. In the affected tissue from our patient, we detected a low signal for CCT5 in the cytoplasm and the nuclei of myofibers, while a stronger signal was present in the sarcolemma and the extracellular matrix. The subunit CCT1, which is known not to be physically linked to CCT5 within the CCT complex, was detected in the patient’s muscle, colocalizing with the CCT5 in the sarcolemmal and subsarcolemmal locations, in the nuclei, and extracellularly, suggesting that the chaperonin complex is formed in the affected myofibers.

*Sarcomeric and extra-sarcomeric proteins.* Immunofluorescence signals for sarcomeric and extra-sarcomeric proteins involved in the maintenance of skeletal muscle tissue were found to be altered in various degrees. Actin, one of the client proteins of the CCT complex, was found in affected myofibers from the patient, but the signal was lower than it was in healthy skeletal muscle. Myosin filaments (MHC IIX) were found in the patient’s muscle, but with weak signal intensity. These findings suggest that the capacity of CCT to fold actin was diminished but still present in the patient. The desmin costameric protein, a muscle-specific type III intermediate filament, was decreased, with altered distribution. The typical banding of the desmin running over the sarcomeric Z-discs was lost in the longitudinal sections, while in the cross sections, desmin was not in the sarcolemma but appeared with an irregular pattern and as intracellular dots in the paracentral and peripheral positions. We found no data in the literature on the details of the interplay between the CCT complex and desmin, but it is known that desmin damage/disturbance/absence are closely related to morphological alterations of skeletal muscle fibers [18,19].

*Other components of the chaperone system (CS).* The CCT chaperonin cooperates directly and indirectly with other chaperones and co-chaperones, such as prefoldin (PFD), Hsp70, and Hsp90, to maintain protein homeostasis [20]. Hsp90 and αB-crystallin play essential roles in the development and maintenance of skeletal muscle tissue, in which they are predominantly localized at Z-discs [21]. In the muscle of our patient, we observed abnormal distribution of Hsp90 and αB-crystallin. The weak IF signal of αB-crystallin appeared as irregular bands in the longitudinal sections and as precipitates/aggregates with desmin in the cross sections. The Hsp90 IF signal appeared in a disorganized pattern, with the absence of the bands observed in healthy skeletal tissue, and colocalized with desmin. These features suggest that the Z-discs of the patient’s myofibers were affected.

*Predicted molecular abnormalities.* Molecular abnormalities of the mutant CCT5 subunit revealed by in silico analysis are listed in Table 3 (illustrative images may be found in references [4,6]).

In silico analysis of the mutant CCT5 molecule, in comparison with the wild-type counterpart, demonstrated that the apical domain was the most affected [6] (Table 3). This is noteworthy because the Leu224Val mutation occurs within the intermediate domain. The apical domain consists of α helices and β sheets: the α helix-10 and a proximal loop (PL) region are necessary to bind the substrate. The α helix-9 acts as a lid to close the chaperoning chamber and an apical loop connects the α helix-10 to the PL, creating highly charged areas to capture the substrates [22]. The apical domain is not directly required for the interactions with other subunits—that role is played by the equatorial domain, which is also involved in ATP hydrolysis [15]. The equatorial domain does not appear to be affected by the mutation, as suggested by our in silico analysis. The intermediate domain acts as a phone cord transmitting to the equatorial domain the message generated by the interactions of the apical domain with the client protein [15]. In this manner, the CCT complex may be able to discriminate between substrates and to induce those conformational changes that are necessary to trigger ATP hydrolysis and protein folding. All these events are finely regulated by chemical and physical conditions of the subunit, the complex, and the folding chamber. Root mean square deviation (RMSD) analysis and heat maps of the mutant molecule showed disturbances of energies in the area close to the position of mutation (position 224) and in the entire apical domain. Molecular dynamics simulations and the radius of gyration (RG) of the entire molecule demonstrated an altered conformation of the apical domain and an opposite distribution of masses of the CCT5 variant, in comparison to the wild-type molecule, in all three examined conditions (Table 3). The mutation Leu224Val in the intermediate domain may induce the conformational alteration of the apical CCT5 domain, resulting in the impairment of its collaboration with the other subunits during substrate recognition, binding, and relocation within the complex chamber. Additional experiments are necessary to determine to what extent and how the CCT complex is formed, including the mutant CCT5 subunit, and to elucidate its functionality. In this regard, existing data suggest that the Leu224Val mutation alters the structure and physicochemical properties of the subunit domains differently than the His147Arg mutation [9].

## 4. Histological and Molecular Abnormalities That Could Be Attributed to Defective Chaperoning

Table 4 presents histological and molecular abnormalities taken from Table 1 and Table 2 that are candidates for investigating if they can be attributed to defective chaperoning by the CCT complex comprising the mutant CCT5 subunit.

The weak IF signal of the CCT5 subunit in diseased skeletal muscle tissue can be caused by transcriptional and translational downregulation of the mutant CCT5 gene and/or by abnormal degradation of the CCT5 mutant protein. The weak signal of the CCT1 subunit might be explained if, as believed, the CCT5 subunit is the starting point for the assembly of the CCT complex that also contains the CCT1 subunit. CCT is a ubiquitously expressed molecular machine that is essential for the folding of various proteins involved in diverse cellular processes, including cell growth, motility, and differentiation [2,24]. Therefore, its malfunction could result in several cell abnormalities. Along this line of thought, one may assume as a working hypothesis that some of the observed abnormalities in the skeletal muscle tissue of our patient occurred because: (1) the CCT complex with the mutant CCT5 was not able to assist the correct folding of cell-cycle proteins, thus contributing to muscle fiber apoptosis; (2) the cytoskeletal proteins were not properly folded and, thereby, caused the altered shapes and sizes of the muscle fibers; (3) the presence of the CCT5 and CCT1 subunits in the extracellular matrix occurred because the CCT complex bearing the mutant CCT5 subunit, with its altered structure, might have been secreted by the muscle fibers—for example, in extracellular vesicles; and (4) the decreased intensity of the Hsp90 signal reflected the reduction of its CCT interactor. The abnormal signal of αB-crystallin, may have been the consequence of the alteration of the skeletal muscle proteome caused by the mutant CCT dysfunction. It remains unclear which molecules modulate the αB-crystallin levels in human skeletal muscle, but it is known that the wild-type CCT complex can suppress the aggregation of human γD-crystallin, whereas the complex containing the CCT5 subunit with the His147Arg mutation is less efficient in suppressing the aggregation of HγD-Crys substrates [25].

## 5. Other Abnormalities

Other abnormalities detected by immunohistochemistry and immunofluorescence that deserve further analysis are presented in Table 5. 

The involvement of Hsp90 has been investigated in Alzheimer’s, Parkinson’s, Huntington’s, and prion diseases [28]. We can hypothesize that at least some of the protein precipitates we found in skeletal muscle may be the result of Hsp90 dysfunction, particularly affecting its role in dealing with protein aggregation. αB-crystallin is the chaperone of desmin; therefore, its decrease or pathologic redistribution in the patient’s muscle may be the cause of the desmin abnormalities we observed. In pathological conditions associated with damage of skeletal muscle, αB-crystallin has been found to be modulating MyoD activity to protect the muscle cells from the pro-apoptotic processes [16]. However, it remains unclear which molecules modulate the αB-crystallin in human skeletal muscle. It is known that the wild-type CCT complex can suppress the aggregation of human γD-crystallin, whereas the CCT complex that contains the CCT5 subunit with the His147Arg mutation is less efficient in suppressing the aggregation of HγD-Crys substrates [25]. Therefore, interactions between CCT and muscular αB-crystallin are worth investigating further, notably in the context of apoptosis. αB-crystallin has different ways of expressing anti-apoptotic potential, including repressing the maturation of procaspase-3 and modulating the Ago2/RISC complex involved in protein homeostasis [16]. A deficiency of αB-crystallin, like that associated with desmin-αB-crystallin precipitated in the skeletal muscle tissue of our patient, might result in the activation of caspase-mediate apoptosis that, because of protein imbalance, may induce the atrophy of skeletal muscle [16].

## 6. Conclusions and Important Issues Deserving Further Research

Some of the findings discussed in this article suggest that certain histological patterns of skeletal muscle tissue associated with the CCT5 Leu224Val mutation are similar to those found in desmin-related myopathies (DRMs) and in α-crystallinopathy. DRMs and αB-Crystallin mutation usually cause severe cardiac dysfunction, e.g., atrial fibrillation, arrhythmogenic left or right ventricular cardiomyopathy, dilated cardiomyopathy, restrictive cardiomyopathy (RCM), and hypertrophic cardiomyopathy [16,29,30]. Although the CCT complex is expressed in cardiac muscle, we did not detect any alteration in heart function or morphology in our patient. However, we did find a diminution of the white matter in the central and peripheral nervous systems in our patient. These findings indicate that non-myelinated nerve fibers innervating heart muscle were not affected. Instead, glial cells, such as oligodendrocytes and Schwann cells, are probably dysfunctional, possibly leading to a disruption of skeletal muscle contraction and to the triggering of the epileptic manifestations registered for the patient. Accordingly, it be assumed, as a working hypothesis, that the apoptosis (and probably also other abnormalities) of the patient’s skeletal muscle cells were caused primarily by the dysfunction of the CCT complex in neurons, rather than in the muscle cells (Figure 1).

Patients carrying the His147Arg mutation in the CCT5 subunit had a demyelinating peripheral neuropathy [3]. Likewise, mutations in the gene encoding Hsp60 were associated with hypomyelinating defects [31]. We could not find reports on the ability of the CCT complex to fold the myelin proteins, but the interplay of CCT with other molecules involved in myelin formation, such as the members of the Hsp70 family, has been proposed [31,32]. Reports about the expression and function of the CCT complex in human myocardium are not available, but the knock-down of cardiac-CCT subunits in the fly, Drosophila melanogaster, model showed impairment in the structure and function of cardiomyocytes [33]. If the human heart CCT had functions similar to those in Drosophila, one may hypothesize that the interactors of the CCT complex differ in both species, so the effects of the mutations in human CCT5 may not affect the heart as they do in the fly model. Further investigations are needed to elucidate whether the apoptosis of skeletal muscle tissue in our patient was the result of (1) a direct injury caused by the defective muscular CCT complex, or (2) an indirect injury caused by the defective neuronal CCT complex, leading to skeletal muscle denervation. A concomitance of different mechanisms might also occur. The use of in vivo gene editing in model systems, for example in zebrafish, may help in evaluating cell viability and in assessing the presence and distribution of proteins of interest (e.g., myelin proteins), and in better understanding the molecular processes in which the CCT5 mutant plays a role in the generation of striated muscle abnormalities.

## Figures and Tables

**Figure 1 biology-12-00641-f001:**
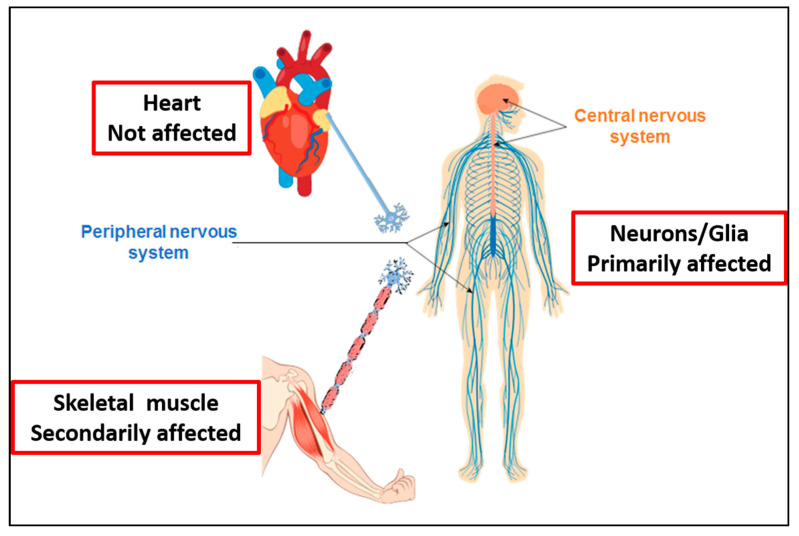
Possible scenario in which the structure, muscles, or nerves may be primarily affected by the Leu224Val mutation in the CCT5 subunit, taking into consideration the absence of clinical or functional alterations of the patient’s heart. We did not find any cardiac alteration in our patient but, instead, we detected a decrease in the white matter in the central and peripheral nervous systems. Consequently, we hypothesize, as a basis for future research, a primary damage of neurons/glia in our patient, causing denervation of the skeletal muscle with its histopathological consequences.

**Table 1 biology-12-00641-t001:** Histomorphological abnormalities in skeletal muscle of a patient with the CCT5 mutant Leu224Val ^1^.

Hematoxylin—Eeosin Staining	Alcian–Pas Staining	TUNEL Assay
Atrophy	Wavy shape of fibers in longitudinal section	Apoptotic fibers
Hypereosinophilia	Swollen nuclei	
Disruption of the tissue architecture	Nuclei in contact with the sarcolemma	
Small fibers in cross-section	Internal paracentric nuclei	
Rounded fibers in cross-section		
Striated pattern inside the fibers in cross section		
Wide inter-fiber space		

^1^ Illustrative images may be found in reference [6].

**Table 2 biology-12-00641-t002:** Molecular abnormalities in skeletal muscle of a patient with the mutant CCT5 Leu224Val ^1^.

Immunofluorescence Plus Confocal Microscopy	Immunohistochemistry
Weak CCT5 signal in the cytoplasm and nucleus, but not as low in the sarcolemma of muscle fibers and in the intercellular matrix	The desmin banding pattern is absent
CCT5 is in the extracellular space (intercellular matrix)	Aggregates containing desmin protein were present at paracentral and peripheral locations within the muscle fibers
CCT5 and CCT1 subunit colocalize mostly in sarcolemma and the extracellular matrix	Nuclei in contact with the sarcolemma
Weak signal of CCT1 in nuclei, closely paralleling the weak signal of CCT5	Internal paracentric nuclei
Myosin signal is marginally increased	
Actin signal is decreased	
Desmin signal decreased, compared with healthy muscle Desmin signal appeared as irregular or trabecular patterns and as dots	
In some fibers, the desmin-labeled sarcolemma was not visible	
Almost no colocalization of CCT5 with desmin was observed in the extracellular matrix of affected tissue	
αB-crystallin signal was weaker than in healthy tissue	
αB-crystallin was present as light bands in longitudinal sections and as precipitates with desmin in the sarcoplasm	
Colocalization of αB-crystallin and desmin was reduced by half in affected muscle, compared with healthy muscle	
Hsp90 was present but appeared with an irregular pattern similar to that of desmin	
Hsp90 and desmin colocalization was reduced in the affected muscle, compared with healthy muscle	

^1^ Illustrative images may be found in reference [6].

**Table 3 biology-12-00641-t003:** Predicted molecular abnormalities ^1^.

Molecular Models under Three Conditions: Nucleotide Free, ATP Bound, and ADP Bound	Radius of Gyration (RG) Versus Time under Three Conditions: Nucleotide Free, ATP Bound, and ADP Bound	Root Mean Square Deviation (RMSD) under Three Conditions: Nucleotide Free, ATP Bound, and ADP Bound	Heat Maps of Wild Type and Mutant CCT5 Subunits under Nucleotide-Free Condition
Apical domain of CCT5 mutant Leu224Val is impaired under all three conditions	Distribution of atomic masses in the CCT5 mutant Leu224Val subunit is totally the opposite as that of the wild-type subunit under all three conditions	Several energy disturbances are present in the apical and intermediate domains of the CCT5 mutant Leu224Val subunit, compared to the wild-type subunit	Instability of the mutant CCT5 subunit is observed in the intermediate domain close to the position of mutation and within the entire apical domain

^1^ Illustrative images may be found in references [6,9].

**Table 4 biology-12-00641-t004:** Histological and molecular abnormalities shown in Table 1 and Table 2 that can be attributed to defective chaperoning by the CCT complex containing the mutant CCT5 subunit.

Abnormality	Hypothesis
Apoptosis	CCT complex may not be able to properly fold cell-cycle proteins
Rounded and small fibers in cross section	CCT complex may not be able to properly fold cytoskeletal proteins, which consequently cannot become well-organized
Weak CCT5 signal in the cytoplasm, the nuclei, and (to a lesser textent) the sarcolemma of muscle fibers	The CCT5 gene may be downregulated, or CCT5 mRNA and/or protein could be degraded at an abnormally increased rate
CCT5 in the extracellular space (intercellular matrix); CCT5 and CCT1 subunits colocalize mostly in sarcolemma and the extracellular matrix	The no-functional mutant CCT5 subunit may be secreted (as subunit or as complex) by the cells via different pathways, including secretion in extracellular vesicles [23]
Weak signal of CCT1 in nuclei, closely paralleling the weak signal of CCT5	CCT5 may be the starting point for the assembling of the CCT complex
Actin signal is decreased	The CCT complex may not be able to properly fold the actin protein
Weak Hsp90 signal	The CCT complex is an interactor of Hsp90 (https://www.picard.ch/Hsp90Int/index.php, accessed on July 2022)
Weak αB-crystallin signal	The CCT5 complex with mutant subunit may alter the proteome of skeletal muscle, inducing the loss of αB-crystallin
Deficient αB-crystallin	CCT/CCT5 complexes may regulate the αB-crystallin in skeletal muscle

**Table 5 biology-12-00641-t005:** Other abnormalities not attributable to defective chaperoning by CCT ^1^.

Abnormalities Probably Caused by Deficient Hsp90	Hypothesis
Precipitates	Hsp90 is involved in regulation of protein misfolding in neurodegenerative diseases [26]
Deficient CCT complex	The CCT complex interacts with Hsp90 (https://www.picard.ch/Hsp90Int/index.php, accessed on July 2022)
**Abnormalities Probably Caused by deficient αB-crystallin**	
Desmin precipitates and deficiency	αB-crystallin is the chaperone of desmin protein; Hsp27 and αB-crystallin (but not Hsp90) have been detected in cytosolic aggregates of a patient affected by desminopathy [27]
General disfunction of skeletal muscle	αB-crystallin can protect skeletal muscle cells from pro-apoptotic effects, such as pathological injuries of skeletal muscle [16]
Apoptosis	αB-crystallin inhibits apoptosis in different ways, e.g., repressing procaspase-3 maturation [16]
Loss of Z-bands	αB-crystallin is highly expressed in Z-discs of adult skeletal muscle [16]
Atrophy	αB-crystallin deficiency results in downregulation of Ago2/RISC, shifting the protein balance of the cell toward atrophy [16]
**Abnormalities Unlikely to Be Caused Directly by Defective Chaperoning**	
General disfunction of skeletal muscle	
Apoptosis	

^1^ These are abnormalities presented in Table 1 and Table 2, but not included in Table 4. Here, two different types of abnormalities are displayed: (1) abnormalities that are probably caused by a deficiency of Hsp90 and/or αB-crystallin, and (2) abnormalities that are unlikely to be caused directly by defective chaperoning.

## Data Availability

Not applicable.

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
