# Peer review of "Histopathology of Skeletal Muscle in a Distal Motor Neuropathy Associated with a Mutant CCT5 Subunit: Clues for Future Developments to Improve Differential Diagnosis and Personalized Therapy"

_biology, 2023, doi:10.3390/biology12050641_

Round 1

Reviewer 1 Report

Overall, this is a good review of the clinical implications of a mutation in CCT5 and the potential mechanisms by which disease may be caused.  I believe that an opening paragraph describing in brief CCT’s structure, allosteric mechanism and clients is necessary.  This background is needed so that the reader can understand that the mutation may have a global allosteric effect on CCT’s function, an effect on binding of CCT5-specific clients and/or on some moonlighting role of CCT5.  Two other comments:

1.      It’s not clear in Table 3 what is meant by ‘energy disturbances’ and ‘instability’.  A more precise description/definition is needed. 

2.     Line 257 – why would the dysfunction be localized to certain cell types? I assume because the relevant clients are expressed there but that should be made clear.

Author Response

We thank the Reviewers for their very useful comments and suggestions, which we have followed in preparing the revised manuscript now submitted. We added explanations and changed various parts of the text, as indicated by the Reviewers. All new or modified parts are visible in Track Changes. Below this message, you will find a point-by-point response to the Reviewers’ comments.

Reviewer 1 comment #1: I believe that an opening paragraph describing in brief CCT’s structure, allosteric mechanism and clients is necessary.  This background is needed so that the reader can understand that the mutation may have a global allosteric effect on CCT’s function, an effect on binding of CCT5-specific clients and/or on some moonlighting role of CCT5.

Authors’ Reply: We thank very much the Reviewer for all the comments and suggestions.

In the revised manuscript now submitted, we added the requested descriptions with the pertinent bibliography (line 57-80).

Reviewer 1 comment #2: It’s not clear in Table 3 what is meant by ‘energy disturbances’ and ‘instability’.  A more precise description/definition is needed.

Authors’ Reply: We thank the Reviewer for this comment; however, these words are related to the Root Mean Square Deviation (RMSD) analysis reported into the bold title of the column of the Table 3.

We think description of the RMSD is not necessary here; moreover, it is a very easy to find descriptions in the literature.

Reviewer 1 comment #3: Line 257 – why would the dysfunction be localized to certain cell types? I assume because the relevant clients are expressed there but that should be made clear.

Authors’ Reply: We thank the Reviewer for this constructive suggestion. We modified the text, accordingly (line 282-288).

We hope the Review is now acceptable for publication in Biology journal.

Sincerely,

Federica Scalia

Reviewer 2 Report

In the review article 'Histopathology of skeletal muscle in a distal motor neuropathy associated with a mutant CCT5 subunit: Clues for future developments to imporve differential diagnosis and personalized therapy' the authors present a mixture of a review article and case report. The topic is interesting. However, this manuscript has severe limitations, which should be fixed before publication:

1.) Please prevent a mixture of review article and case report. This does not make sense. For a review article, please present literature analysis and cite much more literature. For a case report please focus on your specific patient and persent your experiments and clinical investigationsin detail. 

2.) The language needs a strong revision.

3.) Please disucss also cases where other mutations in other chaperone genes cause heart or skeletal muscle diseases: 

Brodehl, A., Gaertner‐Rommel, A., Klauke, B., Grewe, S. A., Schirmer, I., Peterschröder, A., ... & Milting, H. (2017). The novel αB‐crystallin (CRYAB) mutation p. D109G causes restrictive cardiomyopathy. Human mutation38(8), 947-952.

4.) Please write the genes with the official nomenclature and also the mutations with the official nomenclature.

5.) Please present the data in your tables as figures including controls. Perform a statistical analysis.

6.) Please write a full material and method paragraph including statistical analysis.

7.) Please persent all relevant data as figures rather than summarizing tables.

In summary, the manuscript needs a strong revision and rewriting. The authors should focus on a case report rather than trying to write a combination of review article and case report.  

Author Response

We thank the Reviewers for their very useful comments and suggestions, which we have followed in preparing the revised manuscript now submitted. We added explanations and changed various parts of the text, as indicated by the Reviewers. All new or modified parts are visible in Track Changes. Below this message, you will find a point-by-point response to the Reviewers’ comments.

Reviewer 2 comment #1: In the review article 'Histopathology of skeletal muscle in a distal motor neuropathy associated with a mutant CCT5 subunit: Clues for future developments to imporve differential diagnosis and personalized therapy' the authors present a mixture of a review article and case report. The topic is interesting. However, this manuscript has severe limitations, which should be fixed before publication: please prevent a mixture of review article and case report. This does not make sense. For a review article, please present literature analysis and cite much more literature. For a case report please focus on your specific patient and persent your experiments and clinical investigationsin detail.
Authors’ Reply: We thank the Reviewer for all the comments. As we wrote in the “Introduction” the aim of this review is to discuss in detail some of the salient histological and molecular findings we reported in our previous and cited works. We also included the very few related data available in the literature. The objective is to make pathologists and clinicians aware of these disorders, which are likely to be more frequent that now believed. The reported cases of genetic mutation of CCT5 are very few and are all cited in our review.

Reviewer 2 comment #2: The language needs a strong revision.
Authors’ Reply: We checked again the manuscript and English has been carefully reviewed by all authors. The authors have a good knowledge of English, and for one it is the mother tongue. All the typos have been  corrected.

Reviewer 2 comment #3: Please disucss also cases where other mutations in other chaperone genes cause heart or skeletal muscle diseases: Brodehl, A., Gaertner‐Rommel, A., Klauke, B., Grewe, S. A., Schirmer, I., Peterschröder, A., ... & Milting, H. (2017). The novel αB‐crystallin (CRYAB) mutation p. D109G causes restrictive cardiomyopathy. Human mutation, 38(8), 947-952.
Authors’ Reply: We added the references suggested where they fit within the aim of the review (the paper by Brodehl et al., was already cited in the original submission) (line 281). However, the focus of this work is not the possible involvement of all the chaperones in skeletal and heart muscle. The αB‐crystallin has been discussed pertinently to our purposes.

Reviewer 2 comment #4: Please write the genes with the official nomenclature and also the mutations with the official nomenclature.
Authors’ Reply:  We checked all the names indicating the gene (cct5, as in line 113) or the protein of CCT5 subunit (in all other parts of the text we clearly refer to protein, e.g., “CCT5 subunit; CCT5 protein; CCT5 molecule; CCT5-Leu224Val; His147Arg CCT5 mutation”).

Reviewer 2 comment #5: Please present the data in your tables as figures including controls. Perform a statistical analysis.
Reviewer 2 comment #6: Please write a full material and method paragraph including statistical analysis.
Authors’ Reply:  These comments do not apply. Our manuscript is a review.

Reviewer 2 comment #7: Please persent all relevant data as figures rather than summarizing tables.
Authors’ Reply:  This is a Review paper and all the pertinent figures of the related summarizing tables are cited.

Reviewer 2 comment #8: In summary, the manuscript needs a strong revision and rewriting. The authors should focus on a case report rather than trying to write a combination of review article and case report.
Authors’ Reply:  We modified the text and clarified that this paper is a Review with aims described above here and in the text submitted (lone 86-94).

Round 2

Reviewer 2 Report

The authors have addressed some of my points. The editor has to decide if the answers are sufficient. In my view, some points like missing figures are still open. However, I think that the editors should see the complete picture based on all reviewers.